# Peptosome: A New Efficient Transfection Tool as an Alternative to Liposome

**DOI:** 10.3390/ijms25136918

**Published:** 2024-06-24

**Authors:** Maliheh Manteghi, Ozge Can, Tanil Kocagoz

**Affiliations:** 1Department of Medical Biotechnology, Institute of Health Sciences, Acibadem Mehmet Ali Aydinlar University, Istanbul 34752, Turkey; maliheh.manteghi@live.acibadem.edu.tr; 2Department of Biomedical Engineering, Faculty of Engineering and Natural Sciences, Acibadem Mehmet Ali Aydinlar University, Istanbul 34752, Turkey; ozge.can@acibadem.edu.tr; 3Department of Medical Microbiology, School of Medicine, Acibadem Mehmet Ali Aydinlar University, Istanbul 34752, Turkey

**Keywords:** gene therapy, gene delivery, cell-penetrating peptide, histone

## Abstract

Gene therapy is one of the most promising techniques for treating genetic diseases and cancer. The current most important problem in gene therapy is gene delivery. Viral and non-viral vectors like liposomes, used for gene delivery, have many limitations. We have developed new hybrid peptides by combining cell-penetrating peptides (CPPs) with the DNA-binding domain of the human histone H4 protein. These small peptides bind to DNA molecules through their histone domain, leaving the CPP part free and available for binding and penetration into cells, forming complexes that we named “peptosomes”. We evaluated the transfection efficiency of several hybrid peptides by delivering a plasmid carrying the green fluorescent protein gene and following its expression by fluorescent microscopy. Among several hybrid peptides, TM3 achieved a gene delivery efficiency of 76%, compared to 52% for Lipofectamine 2000. TM3 peptosomes may become important gene delivery tools with several advantages over current gene delivery agents.

## 1. Introduction

Humans have long been affected by genetic diseases. After years of research, the genetic basis of many of these diseases has been established. Consequently, over the past two decades, gene therapy, which works by inserting a gene into a patient’s cells, has become an important strategy for preventing or treating genetic disorders and cancer [1,2]. The discovery of the clustered regularly interspaced short palindromic repeats (CRISPR)/Cas9 system was a big advancement in editing genes. This was a major move towards treating genetic diseases [3]. Since Cas9 is a gene-editing enzyme of bacterial origin and is not present in human cells, CRISPR/Cas9 gene modification requires sending a gene into the cell that can express Cas9 together with a guide RNA specific for the region to be modified in the genome. Therefore, the guide RNA sequence and the gene for Cas9 are cloned on a plasmid, which is subsequently delivered into the cells. This necessitates the transfer of larger DNA into the cells [4]. However, among several challenges that need to be solved for efficient gene therapy, gene delivery remains the leading problem [5]. To overcome this problem, several approaches for gene delivery have been developed to efficiently introduce foreign genetic material and ensure its stability within host cells. To ensure effective delivery, the system must not damage cells or induce an inflammatory response in the organism, and the gene of interest must cross multiple barriers during the process [6]. In this regard, the foreign gene must be transported inside the cell nucleus to induce gene expression. The gene must remain stable once inside the cell. The stability of a genetic element means that it can maintain certain characteristic behaviors for long periods without accumulating mutations inside the cell, without interacting with other structures in the host genome, and without undergoing structural, genetic, or epigenetic changes. The gene delivery system must provide appropriate conditions to protect the integrity of the gene [7]. Scientists use different gene delivery tools, including viral vectors and nonviral systems such as mechanical or chemical methods. Viral gene delivery is achieved by using viruses in which disease-causing genes are deleted and the gene needed to treat a genetic disease is added. Retrovirus, adenovirus, and adeno-associated viruses have been widely used because of their high efficiency in gene transfer [8]. Retroviral vectors are produced in host cells that contain viral genes needed for virus replication, which are deleted from the virus to inhibit their replication in the cells to which they deliver the gene for treatment. However, the integrase gene is kept in the viral vector genome to produce an integrase enzyme that integrates the desired gene into the human cell genome by recombination using long terminal repeat (LTR) sequences. This enables long-term expression of the gene and transfers the gene to daughter cells when the cell divides. However, since the site of integration cannot be controlled, there is always concern about insertional mutagenesis, which can inactivate a tumor suppressor gene or create an oncogene that will transform the cell into a cancer cell [9]. The use of viral vectors is associated with potential safety concerns, such as the possibility of an immune response, as well as the limitations of viral vectors [10]. Inserting genetic material into the host genome can sometimes disrupt normal gene function, potentially leading to adverse effects or diseases [11].

Another alternative approach for introducing genes involves employing natural or synthetic substances, commonly referred to as nonviral vectors. These vectors create complexes composed of DNA, proteins, polymers, or lipids organized into particles with the ability to effectively transport genes into cells.

Nonviral gene delivery systems, such as cationic lipid-based methods, which have a positive charge, are easy to prepare and demonstrate high efficiency. However, they may trigger acute immune responses. Non-lipid cationic polymers exhibit high in vitro efficiency but are toxic to cells and display low efficiency in vivo. They are also shown to have the potential to induce immune responses. When lipids are combined with cationic polymers to create hybrid molecules, they demonstrate notable in vitro success but show reduced activity in vivo. Direct needle injection of nucleic acids, a mechanical method, is simple but exhibits low efficiency, confining gene expression to the needle track. Gene gun technology, relying on pressure, achieves good efficiency but limits gene transfer to targeted areas and requires surgical procedures for internal organs. Electroporation, using electric pulses, displays high efficiency but may cause tissue damage and confine gene transfer to specific regions, necessitating surgical interventions for internal organs. Hydrodynamic delivery, utilizing hydrodynamic pressure, is simple and highly efficient, offering site-specificity; however, it requires catheter insertion in large animals [9]. Among nonviral delivery tools, liposomes have become the most used vectors. Liposomes offer high transfection efficiency in vitro, making them ideal for cell-based studies and high-throughput experiments [12]. However, they can induce cytotoxicity or trigger immune responses in cells, especially when used at higher concentrations [13,14]. This cytotoxicity can negatively affect cell viability and overall experimental outcomes. In addition, the lipids that are used to make liposomes are expensive and challenging to produce [15]. Since each method developed so far has its own set of advantages and limitations, further improvements in gene delivery systems are needed.

In recent years, cell-penetrating peptides (CPPs) such as integrin beta fragments have become promising nonviral vectors to deliver various cargoes into cells, including nucleic acids, plasmid DNA, small interfering RNA (siRNA), and antisense oligonucleotides, making them an asset in the development of new therapeutic strategies because of their ability to integrate and cross biological membranes [16]. The processes facilitating cellular internalization through CPPs use a combination of energy-dependent and energy-independent mechanisms. This feature enables the effective delivery of cargo into the cytoplasm and, in certain instances, even into the nucleus of the cell [17]. CPPs are short peptides, typically composed of 5–30 amino acids rich in arginine and lysine amino-acid groups, and can be easily produced at low cost. Their safety has been demonstrated over the last 10 years [12]. Integrin beta fragments exhibit high affinity for cell membranes, primarily through interactions with lipid bilayers. The hydrophobic nature of this peptide allows it to readily associate with the hydrophobic core of cell membranes, facilitating efficient translocation across cellular barriers [18]. CPPs can overcome some of the limitations of viral vectors and liposome-based delivery systems. The properties of CPPs not only allow for effective cellular internalization but also offer potential solutions to challenges encountered with viral vector-based gene delivery systems [19].

CPPs are used successfully to facilitate the delivery of protein-based therapeutics in pathological conditions such as cancer, inflammatory diseases, oxidative stress-related disorders, diabetes, and brain injury [20]. However, protein stability relies on weak non-covalent interactions between secondary, tertiary, and quaternary structures, which must be preserved throughout the delivery process for their activity and their stability to proteases [21]. To improve the efficiency of the delivery of proteins into cells, different types of lipid- and polymer-based vectors, including liposomes, microparticles, and nanoparticles, have been used, but their efficiency has been relatively poor [22]. Increasing the efficiency of CPPs, as well as optimizing their gene delivery features, is an important area of research for the advancement of gene therapy. However, CPPs lack nucleic acid-binding amino acid residues and a nuclear localization signal (NLS) [23].

Cell-penetrating peptides (CPPs) have emerged as promising tools in the field of vaccination too, particularly in the context of mRNA vaccines. mRNA vaccines show great potential in the field of cancer management. It is emphasized that accurately defining the sequence of the target antigen is crucial for the formulation and production of an effective mRNA vaccine [24]. CPPs possess the ability to translocate across cellular membranes, facilitating the intracellular delivery of mRNA. In the context of vaccination, CPPs play a crucial role in enhancing the cellular uptake of mRNA vaccines, leading to improved immunogenicity. 

On the other hand, lipid nanoparticles, including liposomes, are also being extensively studied in the development of mRNA vaccines against infectious diseases and cancer [25]. Liposome-encapsulated mRNA vaccines have shown efficiency in inducing both humoral and cellular immune responses [26]. However, for mRNA vaccines, using CPPs might simplify the formulation process compared to more complex lipid-based systems, potentially reducing production costs [27].

One of the key challenges in vaccine development is the efficient delivery of genetic material to the target cells, especially antigen-presenting cells (APCs) like dendritic cells. CPPs, due to their cell-penetrating properties, aid in overcoming cellular barriers, allowing the mRNA to enter cells more effectively. This enhanced delivery contributes to an increased expression of the encoded antigen, leading to a stronger immune response. A notable example of CPPs in mRNA vaccine technology is the incorporation of arginine-rich peptides. These peptides contain a high proportion of positively charged amino acids, facilitating interaction with negatively charged cell membranes and promoting cellular uptake. Studies have demonstrated the successful use of arginine-rich antisense oligonucleotide CPPs in improving the delivery and efficiency of mRNA vaccines, with enhanced antigen expression and subsequent immune responses [28].

By enhancing cellular uptake and bioavailability, CPPs may enable the use of lower vaccine doses while maintaining or even improving efficiency. This can be particularly advantageous in reducing production costs and mitigating potential side effects associated with higher doses [29]. Also, CPPs can be modified to include targeting sequences, allowing for the specific delivery of mRNA vaccines to certain cell types or tissues. This targeted approach enhances the precision of vaccine delivery and minimizes off-target effects [30].

In this study, we developed a hybrid peptide with CPP residues on one side and a segment of human histone H4 that binds to DNA on the other side and investigated the gene delivery efficiency of this peptide into cells. We hypothesized that the histone segment would bind to DNA, and these peptides would cover a supercoiled plasmid DNA to form a liposome-like structure. The cell-penetrating part of the peptides that will position outside will start endocytosis when they contact a cell (Figure 1).

## 2. Results

### 2.1. Peptide Synthesis and Analysis

All peptides were purified by HPLC. The TM3 peptide was obtained by collecting the eluent from the peak between 21 and 22 min (Figure 2).

### 2.2. Peptosome Formation

The formation of peptosomes when plasmid DNA was mixed with hybrid peptides was analyzed by transmission electron microscopy (TEM) (Figure 3).

TEM images showed the formation of peptosomes as predicted by mixing plasmid DNA with TM3 peptide. This unique structure, consisting of the combination of an integrin beta-3 segment and a nine-amino acid region of histone H4, forms the basis of this gene delivery vector. The histone H4 domain of TM3 facilitates DNA binding, ensuring that the CPP (cell-penetrating peptide) component remains available for effective binding and penetration into cells.

### 2.3. Transfection

Transfection of cells was possible when we used peptides featuring a combination of an integrin beta-3 fragment and a nine-amino acid segment of histone H4 (Table 1).

Although it was possible to transfect the plasmid with TM1 and TM2 into cells, the best transfection efficiency was achieved with TM3 (Patent No: TR2024/000662). The transfection efficiency was better for TM3 than for Lipofectamine at both 24 and 48 h in terms of the percentage of cells transformed (Figure 4).

TM4, which contained only part of histone H4, and TM5, which only contained a fragment of integrin beta 3, did not permit transfection. When we used the growth factor receptor-bound protein SH2 domain in place of integrin beta-3 (TM6), the transfection activity of the peptide was completely lost. In TM7 and TM8, the CPP component of TM3 was modified by changing one or two valine residues to leucine to improve transfection efficiency. However, these changes resulted in the complete loss of transfection activity. Thus, TM3 was identified as the peptide with the best transfection activity. The resulting peptosomes, particularly those formed with the TM3 peptide, exhibited promising transfection abilities. The success of TM3 in transfection can be attributed to the strategic combination of CPPs and the DNA-binding domain of histone H4. The histone domain of TM3 binds to DNA molecules, leaving the CPP part free and available for binding and penetration into cells. This unique structure forms the basis of the peptosomes, ensuring efficient gene delivery. The formation of peptosomes was confirmed through transmission electron microscopy (TEM), providing visual evidence of the complexes formed between TM3 and plasmid DNA.

The transfection efficiency of Lipofectamine 2000 and TM3 at 24 and 48 h in HeLa cells is presented in Table 2.

The percentage of transfected cells expressing GFP increased at 48 h compared to the 24 h evaluation.

The evaluation of hybrid peptides revealed that TM3, featuring a combination of an integrin beta-3 fragment and a nine-amino acid segment of histone H4, exhibited the highest transfection efficiency. TM3 achieved a transfection efficiency of 52% in 24 h and 76% in 48 h. These values were 27% and 52% with Lipofectamine 2000, respectively.

## 3. Discussion

In recent years, proteomics has significantly advanced our understanding of cancer biology and provided new avenues for cancer treatment. Proteomics, the large-scale study of proteins, is essential for understanding the complexities of cancer-causing mechanisms at the molecular level as well as for producing new therapeutic tools. As a result of these studies, anticancer monoclonal antibodies became important biological drugs for cancer treatment [31,32]. Small synthetic peptides, which are much easier to produce than monoclonal antibodies, are also being used more and more to develop new therapeutic regimens. In addition to their direct use for treatment, they may also be used as anticancer drug carriers [33]. In this study, we evaluated their potential for the delivery of genetic material into the cells. The special design of these peptides may enable targeting cancer cells.

Gene therapy has emerged as a promising method for treating genetic diseases, cancer, autoimmune diseases, and infectious diseases that are otherwise incurable [15,34]. The most important problem for gene therapy is the lack of an effective and safe gene delivery system that can be used both in vitro and in vivo. DNA and RNA vaccines, which require gene delivery, have recently become important tools for protecting against infections and treating cancer [35,36].

Currently, researchers use a variety of methods to deliver genes, including both viral vectors and nonviral transfection agents. The use of viral vectors and liposomes has been a common strategy, but limitations and safety concerns have led researchers to explore alternative delivery methods. Viral vectors such as retroviruses, adenoviruses, and adeno-associated viruses that are extensively utilized in gene therapy have gained popularity because of their effectiveness in gene transfer. However, the use of viral vectors poses potential safety issues, including the risk of immune reactions, in addition to the limitations associated with these vectors. Immune responses triggered by viral vectors may limit their therapeutic applications. For example, retroviruses utilize an integrase enzyme to integrate the desired gene into the human cell genome, enabling long-term expression. However, concerns arise regarding insertional mutagenesis, wherein the site of integration cannot be controlled, potentially leading to adverse effects such as the inactivation of tumor suppressor genes or the creation of oncogenes that could transform cells into cancer cells [10]. Therefore, alternative transfection tools such as liposomes, nanoparticles, peptides, and techniques using naked DNA have emerged as safer and more economical techniques. Some of these tools have less toxicity and lower potential to trigger immune responses, making them advantageous in terms of scalability [15].

Liposomes, as nonviral vectors, are widely used in delivery systems. They can encapsulate nucleic acids, offering flexibility in cargo selection, but they have some disadvantages, such as difficult production, high costs, and limited penetration. Whereas liposomes can exhibit cargo leakage during circulation, CPPs offer improved cargo protection, minimizing premature release and enhancing drug stability [37].

Even so, the application of CPPs is constrained by several notable limitations. Firstly, CPPs often exhibit non-specific uptake, leading to off-target effects and potential toxicity in non-target cells. Additionally, the efficiency of CPP-mediated delivery can be highly variable depending on the type of cargo, the cell type, and the specific CPP used. Another significant limitation is the potential for rapid degradation by extracellular proteases, reducing the effective concentration of CPPs available for cell entry. Furthermore, the endosomal entrapment of CPP-cargo complexes can hinder the release of the therapeutic agents into the cytosol, thereby diminishing their intended biological activity. These challenges underscore the need for further optimization and development of CPPs to enhance their specificity, stability, and delivery efficiency for therapeutic applications [16]. After considering all this, among all gene delivery agents, CPPs have gained attention in recent years as nonviral vectors that can efficiently deliver various cargoes into cells. In most of the research involving CPPs, peptides such as penetratin, Tat, and polyarginine peptides were initially used for translocation purposes [38,39,40]. Because of their advantages, such as stability in extreme temperatures and easy production, CPPs emerged as promising tools for gene delivery [41]. CPPs generally exhibit low immunogenicity and cytotoxicity compared to some liposomal formulations [42]. These characteristics are crucial for the development of safe and effective drug delivery systems. CPPs can also be easily modified via chemical conjugation with different molecules to optimize their properties, including stability and specificity. Thereby, CPPs can be tailored to meet the requirements of specific drug and gene delivery applications [43,44].

El-Andaloussi et al. [45] developed a CPP named M918 for the efficient delivery of proteins and peptides into cells. The results illustrated that M918 exhibits remarkable cell-penetrating properties, displaying high efficiency and stability in delivering a wide range of proteins and peptides into different cell types. The peptide displayed efficient internalization and no significant cytotoxicity, making it a safe and viable option for cellular delivery applications. In another study, Kevin et al. validated CPPs as potential molecules for improving the cytosolic delivery of therapeutic oligonucleotides in the liver, but safety remained a concern for this approach [46]. A study by Kato et al. indicated that the design of novel CPPs using unnatural amino acids for plasmid DNA delivery was possible. They found that prolonged post-incubation led to significantly higher transfection efficiencies [47].

This research presents a novel, easy-to-produce peptide with low cell toxicity as a highly efficient gene delivery tool. The novelty of our gene delivery agent is that it is a hybrid peptide with CPP residues on one side and a human histone H4 segment that binds to DNA on the other side. We designed several peptides by combining different peptide sequences derived from different CPPs with the DNA-binding component of the histone H4 protein. The H4 segment cannot deliver genes when used without a leading CPP sequence. Among our hybrid peptides, the most efficient gene delivery vector was TM3 (Table 1). The binding of the histone component of TM3 to DNA positions their free CPP component outside the “peptosome,” making them available for cell binding and penetration (Figure 3). TEM images of peptosomes indicated spherical formations with two different densities: a thinner layer covering a more massive region in the middle, as can be expected by the free CPP part of TM3 outside, and the histone part bound to the surface of supercoiled plasmid DNA inside.

TM3 was identified as the peptide with the best transfection activity. The resulting peptosomes, particularly those formed with the TM3 peptide, exhibited promising transfection abilities. The success of TM3 in transfection can be attributed to the strategic combination of CPPs and the DNA-binding domain of histone H4. The histone domain of TM3 binds to DNA molecules, leaving the CPP part free and available for binding and penetration into cells. This unique structure forms the basis of the peptosomes, ensuring efficient gene delivery.

The CPP sequence was an important factor in the transfection efficiency of peptosomes. In the transfection, we did not include any CPP part, as in TM4, which contained only part of histone H4, the transfection feature of the peptide was completely lost. A small portion, including just three amino acids of the integrin beta fragment used in TM3, enabled the highest transfection efficiency. Although transfection was possible when the whole integrin beta was used in TM1 and a larger part in TM2 compared to TM3, their transfection efficiency was much lower than in TM3. TM5, which only contained a fragment of integrin beta 3, may permit transfection; however, it did not enable the expression of GFP. This may be due to the lack of histone sequences, which function as an NLS, directing DNA into the nucleus and enabling the transcription and expression of the transfected gene. When the growth factor receptor-bound protein SH2 domain is used in place of integrin beta-3 in TM6, the transfection activity of the peptide is completely lost. In TM7 and TM8, the CPP component of TM3 was modified by changing one or two valine residues to leucine to improve transfection efficiency. However, these changes resulted in the complete loss of transfection activity. All these results showed that the CPP sequence used in peptides to form peptosomes is very critical.

TM3’s effectiveness in transfection, coupled with its cost-effective synthesis and demonstrated safety of CPPs over the last decade, positions it as a promising candidate for further development.

Exploring the variety of TM3 in delivering different types of genes and cargoes could enhance its applicability across a broader spectrum of gene therapy applications.

Our findings shown in Table 2 demonstrate a notable increase in the percentage of transfected cells expressing GFP at 48 h compared to the 24 h evaluation for both reagents, indicating a time-dependent enhancement of transfection. This observation suggests that longer incubation times might be advantageous for achieving higher transfection rates using these reagents. Moreover, our analysis of hybrid peptides revealed that TM3 exhibited superior transfection efficiency compared to Lipofectamine 2000. Specifically, TM3 achieved transfection efficiencies of 52% and 76% at 24 and 48 h, respectively, whereas Lipofectamine 2000 yielded lower efficiencies of 27% and 52% for the same time points.

These results indicate the potential of TM3 as a promising transfection agent (Figure 4), surpassing the performance of the widely used Lipofectamine 2000. The enhanced transfection efficiency of TM3 could be attributed to its unique composition, which combines elements that facilitate cellular uptake and nuclear localization. Further investigation is warranted to elucidate the underlying mechanisms driving the superior performance of TM3 and to optimize its delivery system for various applications in molecular biology and biotechnology. Overall, our study underscores the importance of exploring novel transfection agents and highlights TM3 as a promising candidate for future research and applications in gene delivery. The application of peptosomes in cancer treatment, particularly for delivering therapeutic genes or CRISPR/Cas9 components, may offer targeted and effective strategies for combating cancer at the genetic level. Considering the relevance of gene therapy in vaccine development, peptosomes could play a role in enhancing the gene delivery of DNA or RNA vaccines.

This study’s findings contribute to addressing the challenges associated with current gene delivery methods and provide a novel approach that could revolutionize the field of gene therapy. Further trials, including animal studies, are needed to evaluate the transfection efficiency and toxicity in vitro and in vivo and develop TM3, which has several advantages over current gene delivery agents, as a gene delivery tool. One of the limitations of this study was the lack of a comparison of TM3 transfection efficiency with previously reported CPPs as transfection agents. This should be included in further studies of the evaluation of TM3. The cost of small peptide production by a peptide synthesizer is much lower compared to the production of lipids used to produce liposomes. The extension of CPPs by histone sequence H4 is not expected to increase considerably. The production of small peptides is much easier and faster compared to liposome lipids, increasing their feasibility. The development of TM3 and the concept of peptosomes may open new avenues for advancing gene delivery systems and may bring us one step closer to overcoming the challenges of treating genetic diseases and cancer through gene therapy.

The implications of these findings are profound for the field of cancer proteomics. Proteomic analysis can provide insights into the molecular mechanisms by which TM3 enhances transfection efficiency, thereby facilitating the optimization of peptide-based delivery systems for cancer therapy. Moreover, understanding the interactions between TM3 and cellular proteins could lead to identifying new therapeutic targets and developing more effective cancer treatments.

### 3.1. Solved Problems in Established Methods

Traditional gene delivery methods, including viral vectors and non-viral systems like liposomes, face several limitations. Viral vectors are highly efficient but carry risks of immune responses and insertional mutagenesis, which can lead to adverse effects such as the creation of oncogenes. Non-viral systems like liposomes, while safer, often result in cytotoxicity and are expensive to produce. The new peptosome system addresses these issues by offering high transfection efficiency with lower toxicity and production costs.

### 3.2. Limitations

Despite the promising results, this study has some limitations. The long-term stability and potential immunogenicity of the peptosomes in vivo have not been thoroughly investigated. Additionally, the study primarily focuses on in vitro transfection efficiency, and further research is needed to evaluate the performance and safety of peptosomes in animal models and clinical settings.

### 3.3. Perspectives

Future research should aim to explore the in vivo applications of peptosomes, assessing their stability, immunogenicity, and overall safety in animal models. Additionally, optimizing the peptide design could further enhance transfection efficiency and specificity. The peptosome technology holds potential for various therapeutic applications, including gene therapy and vaccine technology, making it a promising alternative to current gene delivery systems.

## 4. Materials and Methods

TM1 to TM8 were synthesized by solid-phase peptide synthesis using CEM, Liberty™ Blue, and CEM Discover™, (Matthews, NC, USA). CEM’s Razor™, (Matthews, NC, USA). The American Chemical Society (ACS)-grade chemicals used for peptide synthesis and purification, amino acids, anhydrous dimethylformamide (DMF), diisopropyl carbodiimide (DIC), trifluoroacetic acid (TFA), Oxyma^®^, piperidine, dichloromethane (DCM), triisopropyl silane (TIS), acetonitrile (ACN), diethyl ether, toluene, amyl acetate, ethyl alcohol, and propylene were purchased from Sigma-Aldrich, (St. Louis, MO, USA).

For in vitro experiments, Luria-Bertani Agar culture medium (LBA, Sigma-Aldrich), Luria-Bertani broth culture medium (LBB, Sigma-Aldrich), Tris Acetate EDTA buffer (TAE, Sigma-Aldrich), dimethyl sulfoxide (DMSO, Sigma-Aldrich), and tannic acid solution (Uranyless) were used. Escherichia coli DH5-alpha competent cells (Thermo Fisher Scientific, (Waltham, MA, USA) were used for the production and isolation of plasmids. The HeLa cells (CCL-2™-ATCC) were cultured at 37 °C with 5% CO_2_ in Dulbecco’s modified Eagle’s medium (DMEM, Gibco™, (Grand Island, NE, USA) supplemented with 10% fetal bovine serum (FBS, ATCC 30-2020™ (ATCC, Manassas, CA, USA)) and penicillin-streptomycin (Pen/Strep, Gibco™). OPTI-MEM (Gibco™) was used to obtain a reduced environment for the optimal transfection conditions. Lipofectamine™ 2000 Transfection Reagent (Thermo Fisher Scientific) was used as a negative control.

A fluorescence microscope, ZEISS (Jena, Germany) was used to observe the GFP-expressing cells, and transmission electron microscopy (TEM-Thermo Fisher Scientific TALOS L 120C) was used to visualize the peptosomes.

### 4.1. Design and Characterization of Peptides

Eight peptides named TM1 to TM8 were designed by combining distinct amino acid residues from various CPPs with part of the histone H4 protein, which is rich in positively charged amino acids that bind to DNA (Table 1). The sequences written in bold were taken from the histone H4 protein. TM1 contains a fragment of integrin beta-3 bound to H4. TM2 and TM3 contain different, shorter parts of beta-3 integrin. TM4 consists only of a part of histone H4 without a leading CPP. TM5 consists of a fragment of integrin beta-3 without a histone H4 component. TM6 uses the growth factor receptor-bound protein SH2 domain as a leading CPP. TM7 and TM8 are different versions of TM3, featuring changes in the CPP-leading region.

### 4.2. Synthesis and Purification of TM Peptides

The designed peptides were synthesized using a peptide synthesizer, Liberty Blue™ and discover™, CEM (USA) following the standard fluorenyl methoxycarbonyl protocol on rink amide resin with a loading capacity of 0.7 mmol/g.

Briefly, synthesis was performed using a predetermined scale of 0.10 mmol. The resin was swelled via a 30-min immersion in dimethylformamide (DMF). All amino acids were maintained at a concentration of 0.2 M in DMF to make them adopt the L-conformation. Both the amino acids and resin contained protective groups to prevent unwanted side reactions during synthesis. The first amino acid was bound to the resin by its C-terminus after removing the resin’s functional protection group. Piperidine (20%) was used for deprotection, and 1.0 M oxyma and 0.5 M diisopropyl carbodiimide were used for coupling. The extension of the peptide was achieved by repeating the cycles. After the final N-terminal deprotection, the peptides were released from the resin using a peptide cleavage system (Razor™, CEM via incubation for 35 min at 37 °C with 5 mL of trifluoroacetic acid, triisopropylsilane, and ddH2O at a ratio of 9.5:2.5:2.5 (*v*/*v*/*v*)). To remove solvents, the peptides were extracted three times with cold (−20 °C) diethyl ether. Finally, the peptides were purified by reverse-phase high-performance liquid chromatography RP-HPLC (260 Infinity Quaternary Liquid Chromatography systems, Agilent Technologies, (Santa Clara, CA, USA) C-18 column (RPC 250 × 10 mm ID hydrophobic 6 µm, Agilent VariTide (Santa Clara, CA, USA)) using an appropriate 5–80% ACN (0.025% TFA)/ddH2O (0.05% TFA) gradient.

### 4.3. Preparation of TM Peptide/Plasmid DNA Complexes

pcDNA3-EGFP, which expresses green fluorescent protein (Enhanced GFP) when introduced into cells, was used to investigate the efficiency of TM peptides in the cellular delivery of DNA molecules. Each TM peptide was dissolved separately in deionized water, and several 10-fold dilutions were prepared to obtain final concentrations of 2–2 × 10^5^ pM. The final concentration of the plasmid was 1.3 pM.

In this experiment, the amount of plasmid was kept constant at a final concentration of 1.3 pM. After 5 mg of all peptides were taken separately and dissolved in 1 mL of pure water, the transfection efficiency of different dilutions up to 10^−6^ dilutions as a fraction of cells expressing GFP was examined by fluorescence microscopy. The transfection efficiency for the best dilution of each peptide was tested by using different volumes of 10, 20, 50, 100, and 200 µL of each dilution in the mixture. Each peptide was tested three times to ensure reliability and reproducibility. Upon repeated experimentation, it was observed that TM3 consistently demonstrated superior transfection efficiency compared to other peptides tested.

### 4.4. TEM Imaging of Peptosomes

Each sample (20 µL) was deposited on parafilm, a grid was placed on it, and we waited for 2–3 h until it was dry. Then, to create contrast, one drop of tannic acid solution (Uranyless) was dropped on the sample. After waiting for 5 min, the samples were washed by dipping them 20 times in distilled water in three separate beakers and left to dry for one night. Peptosomes, along with TM3 and pcDNA3-EGF, were imaged by transmission electron microscopy (TEM; Thermo Fisher Scientific TALOS L 120C) (Figure 3).

### 4.5. Transfection

Firstly, HeLa cells (CCL-2™, ATCC) were seeded in 24-well culture plates (45 × 10^4^ cells/well) and incubated overnight in 1 mL of Dulbecco’s modified Eagle’s medium (Gibco™) containing 10% fetal bovine serum (FBS, ATCC 30-2020™) and 1% penicillin/streptomycin stock solution at 37 °C (Pen/Strep, Gibco™). The following day, the medium was aspirated, and the cells were carefully washed two times with PBS 1X at 24-h intervals. The next day, the cells were used when they reached 85–90% confluency.

To optimize transfection efficiency with TM peptides, we prepared mixtures of TM peptides and pcDNA3-EGFP with different ratios and concentrations. TM peptide/plasmid DNA mixtures were incubated for 15 min at room temperature. Subsequently, 100 μL of this mixture was mixed with 900 µL of optiMEM (Gibco™) medium containing 10% FBS and 1% Pen/Strep in each well. OPTI-MEM (Gibco) was used to obtain a reduced environment for the optimal transfection conditions. The best transfection efficiency was obtained with a final concentration of pcDNA3-EGFP at 1.3 pM and TM3 peptide at 200 pM, making a ratio of 154 TM3 molecules per plasmid. Naked plasmid DNA, HeLa cells, and commercially available Lipofectamine™ 2000 Transfection Reagent (Thermo Fisher Scientific)/plasmid DNA mixtures were used as negative controls.

### 4.6. Image Processing and Analysis of Cells

The cells were observed under a fluorescence microscope with a GFP filter at 24 and 48 h with excitation at 470 nm and emission at 525 nm (ZEISS, Germany) to investigate the transformation of cells and the expression of GFP. Transfection efficiency was evaluated by comparing the fluorescence image of the cells with their brightfield counterpart, as seen in Figure 4. Transfection efficiency was calculated by dividing the number of cells yielding green fluorescence signals expressing GFP by the total number of cells in the corresponding brightfield image.

## 5. Patents

Patent No: TR2024/000662.

## Figures and Tables

**Figure 1 ijms-25-06918-f001:**
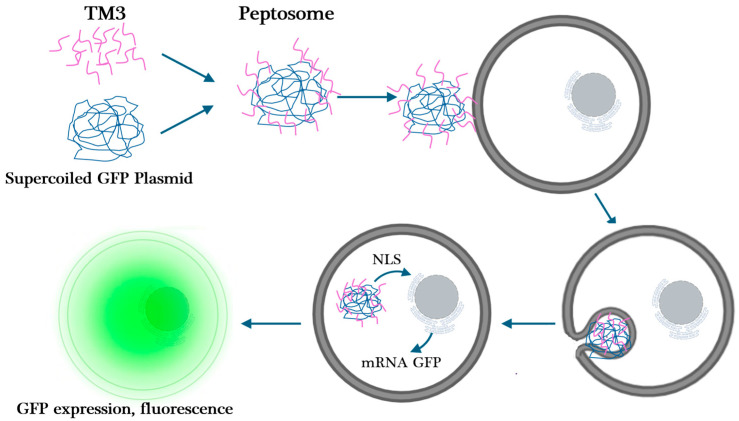
The hypothesized mechanism of peptosome formation. The delivery of DNA into the cell and then to the nucleus.

**Figure 2 ijms-25-06918-f002:**
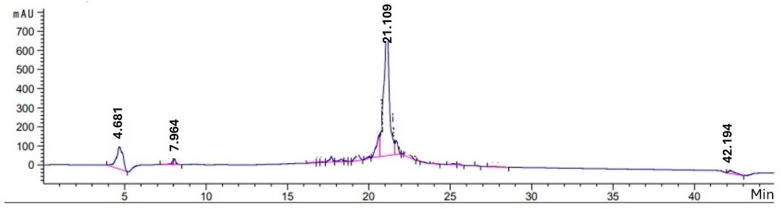
HPLC chromatograms of the TM3 peptide. The eluent, referring to the peak between 21 and 22 min, was collected to obtain the TM3 peptide, which is used in transfection experiments.

**Figure 3 ijms-25-06918-f003:**
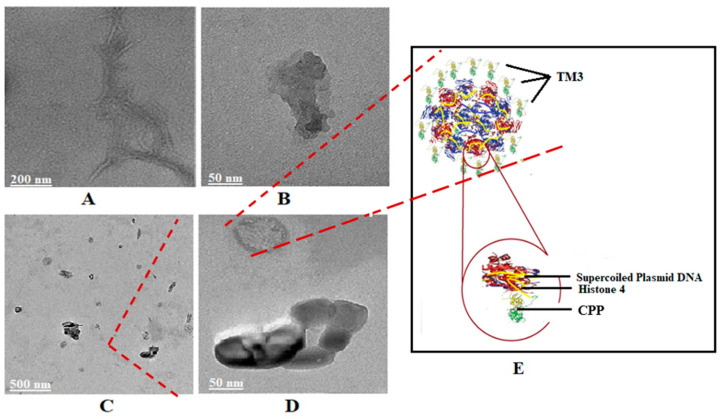
TEM (transmission electron microscopy) image and predicted structure of peptosomes. (**A**) Probable bundles of DNA strands; (**B**) Probable aggregates of TM3 peptide; (**C**) Probable supercoiled plasmid DNA and TM3 peptide complexes (peptosomes), which are about 50 to 100 nm in size; (**D**) A group of peptosomes in higher magnification; (**E**) Predicted structure of the peptosome as supercoiled plasmid DNA surrounded by TM3 peptides.

**Figure 4 ijms-25-06918-f004:**
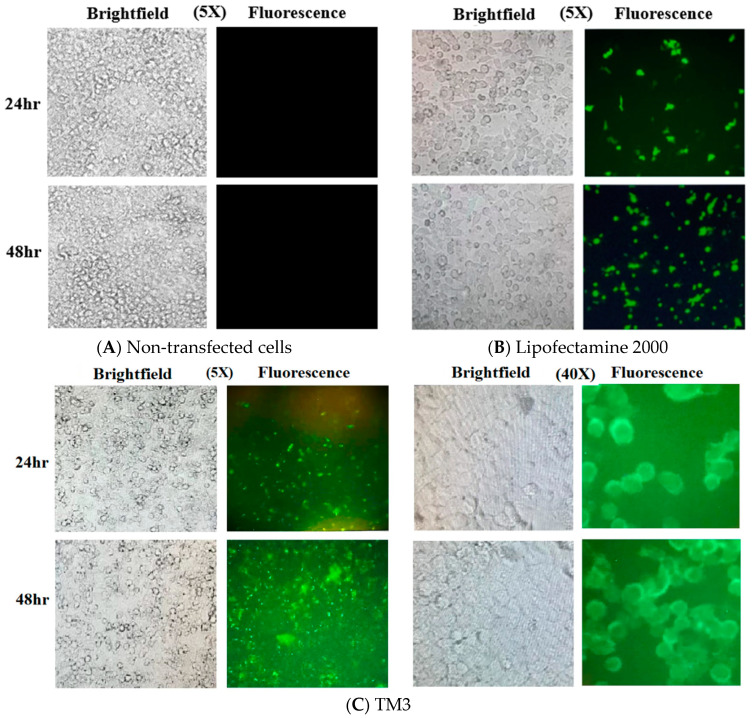
GFP expression (seen as green fluorescence) in HeLa cells: Non-transfected cells (**A**) transfected by Lipofectamine 2000 (**B**) and TM3 (**C**) at 24 and 48 h. The pictures on the brightfields on the left are overlays of areas of the fluorescent pictures on the right. Lipofectamine has better transfection efficiency than TM6 but less efficiency than TM3.

**Table 1 ijms-25-06918-t001:** Peptides are used to form peptosomes. The sequences written in bold represent the histone H4 protein part.

TM Peptides	Sequences
TM1	VTVLAGALAGVGV**GKGGAKRHR**
TM2	VTVLAG**GKGGAKRHR**
TM3	VGV**GKGGAKRHR**
TM4	**GKGGAKRHR**
TM5	VTVLAGALAGVGV
TM6	AAVLLPVLLAAP**GKGGAKRHR**
TM7	LGL**GKGGAKRHR**
TM8	LGV**GKGGAKRHR**

**Table 2 ijms-25-06918-t002:** Lipofectamine 2000 and TM peptides transfection efficiency in 24 h and 48 h in HeLa cells. TM3 showed the highest transfection efficiency at 76%, compared to 52% of Lipofectamine 2000.

Reagent/Time	24 h (%)	48 h (%)
Lipofectamine 2000	27	52
TM1	12	16
TM2	15	18
**TM3**	**52**	**76**
TM4	0	0
TM5	0	0
TM6	0	0
TM7	0	0
TM8	0	0

## Data Availability

Data is contained within the article.

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
