# Peer review of "Peptosome: A New Efficient Transfection Tool as an Alternative to Liposome"

_ijms, 2024, doi:10.3390/ijms25136918_

Round 1

Reviewer 1 Report

Comments and Suggestions for Authors

In this report the authors introduce a new transfection tool which the call “peptosome”. They use several histone H4 peptides to do this and conclude that one of them (TM3) is the best. The data has value and warrants publication. However, the way the paper is structured is somewhat confusing. Below I provide some ideas for revisions.

Revisions

1.       It would be best to include a cartoon figure showing the components of these peptosomes. As it stands now it is a bit confusing even to me and possibly more for a reader not familiar with these experiments. For example, Figure 2E shows H4, DNA, and CPP but where is the GFP coming from in Figure 3. Clearly, this is testing efficiency, but it is confusing as it is laid out now. Please see also point 4 below on efficiency. An introductory cartoon diagram may be able to address both this point and point 4.

2.       Sections 3.1 and 3.2 only mention TM3 and only data for TM3 is shown. Do the other peptides not form aggregates? Why are they not mentioned in these sections? Data should be provided (as supplemental) for all peptides or at least it should be discussed what the other peptides look like. They are addressed later (see my point 4 below) but only briefly.

3.       Line 235 mentions green lines in figure 1 but I only see blue and pink lines. Please correct.

4.       Section 3.3 again mentions TM3 as having the best transfection efficiency with discussion on the other peptides as not doing as well. How was transfection efficiency assessed? I cannot tell from section 2.5 in the methods. Have GFP statistics been done (e.g. percent of GFP cells)? If so, please provide in supplemental data. Table 2 shows only TM3 and some percentages but what are those percentages? GFP expression? The text mentions percentage of transfected cells and I assume that this was assessed using GFP fluorescence. Please be more specific in the legend.

5.       Figure 3 is not well labeled. If you use capital letters (e.g. A, B, C, etc) for figure 2, please do the same for subsequent figures. Figure 3 uses lower case letters.  

6.       The discussion spends too much time rehashing introduction information (lines 279-331). This section should get to the point faster. This happens at paragraph starting at line 332 but even so it is not clear what it is about the TM3 peptide that makes it the best candidate? The authors should discuss the differences between the peptides in more detail and address/speculate about potential characteristics that make TM3 ideal.

Reviewer 2 Report

Comments and Suggestions for Authors

Authors propose a new method of gene delivery via peptosome. It’s a promising tool for multiple purposes. However, the study is too preliminary for publication as an article. 

The idea of using cell-penetrating peptides (CPP) as a gene delivery tool is not new. An comprehensive review can be found at PMID: 32138146. In current manuscript authors proposed to combine CPP with histone H4 peptides. Is this strategy comparable or superior to previous methods, in terms of efficiency, cost or feasibility?

Figure 2 shows that the size of a typical peptosome is 50~100nm, which is apparently smaller than naked plasmid DNA or TM3 peptide aggregates. Could authors please provide further evidence of peptosome structure?

Figure 3 should show an overlay of bright field and GFP to show that the fluorescence is localized intracellularly. A non-transfected (or DNA only) control should be included as well.

Introduction and discussion sections are not sufficiently supported by the results. Authors could, for example, include more data comparing different TM peptides with or without CPP to characterize the new system, or delivery of other target genes, RNAi, or CRISPR/Cas9 vectors to reinforce its application potential. 

Comments on the Quality of English Language

Authors could re-organize the manuscript to make it more concise.

Reviewer 3 Report

Comments and Suggestions for Authors

IJMS is a renowned journal and this work looks incomplete. So, I reject this manuscript to be published in IJMS for the following reasons.

·        Title of the manuscript is very general.

·        The introduction part is interesting and it has a logical flow. It describes the other technology for gene therapy also. Authors have described many advantages with recent research of CPPs. But authors haven’t described the limitations of CPPs and in which specific conditions its better to use the alternative tools for gene therapy.

·        Poor presentation of Figure 1.

·        Figure 2e can be improved.

·        How author conclude the predicted structure of peptosome?

·        The reader cannot know the average size of peptosome.

·        No advance study is done by the authors.

·        The style of the references is not consistent.

Comments on the Quality of English Language

Proof reading is required for results and discussion part.

Round 2

Reviewer 1 Report

Comments and Suggestions for Authors

The authors have made the changes I requested. This reviewer is satisfied.

Author Response

The authors have made the changes I requested. This reviewer is satisfied.

Reviewer 2 Report

Comments and Suggestions for Authors

Authors have addressed my major concerns and improved the quality of manuscript.

Comments on the Quality of English Language

Authors may consider a more consice discussion section, including a summary of current study, solved problems in established methods, limitations and perspectives.

Reviewer 3 Report

Comments and Suggestions for Authors

Revised manuscript is acceptable in this present form.

Author Response

Revised manuscript is acceptable in this present form.